# REWARDLESS OPEN-ENDED LEARNING (ROEL)

## ABSTRACT

Open-ended learning algorithms aim to automatically generate challenges and solutions to an unending sequence of learning opportunities. In Reinforcement Learning (RL) recent approaches to open-ended learning, such as Paired Open-Ended Trailblazer (POET), focus on collecting a diverse set of solutions based on the novelty of an agents pre-defined reward function. In many practical RL tasks defining an effective reward function a priori is often hard and can hinder an agents ability to explore many behaviors that could ultimately be more performant. In this work we combine open-ended learning with unsupervised reinforcement learning to train agents to learn a diverse set of complex skills. We propose a procedure to combine skill-discovery via mutual information, using the POET algorithm as an open-ended framework to teach agents increasingly complex groups of diverse skills. Experimentally we demonstrate this approach yields agents capable of demonstrating identifiable skills over a range of environments, that can be extracted and utilized to solve a variety of tasks.

## 1 INTRODUCTION

Development of machine learning over the past several decades has principally relied upon researchers solving a series of challenges proposed by experts within the community (e.g. MNIST Lecun et al. (1998), ATARI Bellemare et al. (2013) and robotics Yu et al. (2021)). Learning algorithms typically solve these challenges through a huge number of observations, and whilst they have exceeded human-level performance on a variety of these tasks (Silver et al., 2017), (Berner et al., 2019), they often require significantly more examples before coming close to matching human performance (Toromanoff et al., 2019). Humans in contrast leverage many diverse experiences to solve complex tasks (Wang et al., 2018) often without a well defined objective or reward. Should we attempt to teach reinforcement learning (RL) agents in a similar fashion? We use this question to motivate the design a procedure that aims to learn a diverse set of increasingly complex novel behaviors without supervision (labels) or incentives (external rewards).

Open-ended algorithms aim to automatically generate challenges to be solved (Forestier et al., 2020), (Schmidhuber, 2012), (Standish, 2002), (Secretan et al., 2011) that can serve as stepping stones to increasingly complex problems. These algorithms are useful in designing a general unsupervised learning procedure because they do not rely upon manually selected challenges; instead they search for challenges that form stepping stones to solve larger ones, collecting a diverse set of experiences along the way. Designing algorithms to achieve these open-endedness can be challenging. Often requiring a careful balance between diversity, optimization (Mouret & Clune, 2015), (Pugh et al., 2016), and a domain complex enough for sufficient experiences/challenges to be generated in order to learn a suitable curricula.

The Paired Open-Ended Trailblazer (POET) algorithm (Wang et al., 2019), (Wang et al., 2020) is a recent example of an attempt to solve these challenges in the context of supervised RL. POET uses the Minimal Criterion Coevolution (MCC) algorithm (Brant & Stanley, 2017) to co-evolve increasingly complex environments and agents and then directly optimizes an RL objective either with evolution strategies (ES) (Salimans et al., 2017) or proximal policy optimization (PPO) (Schulman et al., 2017). Within POET both the MCC for suitable learning outcomes and agent optimization rely upon the extrinsic reward within the environment. This approach works well for the reward-dense bipedal walker control problem (Brockman et al., 2016) but can often become challenging, even in conceivably simple RL problems, to generate desired outcomes (Vecerik et al., 2018).

This problem becomes more challenging when the reward is sparse, especially with a long-horizon, where the agent effectively has no reward until termination. Reward-shaping can be used (Ng et al., 1999) to help improve the density of reward and aide exploration. However, reward-shaping does not encourage open-ended learning, because it rewards the agent for following a hand-designed objective driven path, which limits open-endedness and often leads to poor performance (Lehman & Stanley, 2011b). An alternate approach is found in the field of unsupervised RL where we don't use an extrinsic reward function at all. Instead the objective is to learn a group of behaviors and hope to observe the agent solving task through one of our explorations, an approach that shares some characteristics with the problem of novelty search (NS) in open-ended learning (Lehman & Stanley, 2011a).

The acquisition of a diverse set of *useful* skills without extrinsic reward is in itself no-easy task. We consider a *skill* as a latent-variable conditioned policy that alters the state of the environment consistently (Gregor et al., 2016), (Eysenbach et al., 2018). The utility of a skill is typically based on its diversity to another, for example whilst one skill might be standing still, walking, dancing and running are all vastly different and a good algorithm will aim to learn the entire set of skills. Skills are then often selected for a task specific problem either by imitation learning or by directly selecting an encoding on the latent-variable conditioned policy. The learned behaviors can also be used as a pre-conditioned policy for regular RL with rewards similar to pre-training on ImageNet (Deng et al., 2009)

In this paper we propose Rewardless Open-Ended Learning (ROEL) an unsupervised deep RL algorithm based upon the POET framework. The key-idea of this work is to use constant environment mutation to generate a diverse set of skills capable of solving increasingly complex and novel problems. We hypothesize this combination of open-ended learning with unsupervised RL will generate sets of behaviors that are more diverse and general than unsupervised RL agents simply trained on a static environment. Behaviors can then be selected for a specific task by directly selecting from a latent encoding, or by comparing the state-space transition to a demonstration. Then, if a skill can only be demoed on a single environment, our environment generalist approach ensures this type of behavior will generalize across many environments.

Our paper makes the following contributions:

- We present a new algorithm, ROEL, that combines unsupervised RL with open-ended learning to generate diverse and general behaviors (Section 3). An approach that has not been attempted before with agent-environment co-evolution (Section 2).

- We train an unsupervised learning algorithm on the bipedal-walker environment. An environment with a difficult task objective and a non-stationary environment transition.

- We prove empirically, that our approach is capable of learning more general and diverse groups of policies than single environments, especially when deployed to environments with dynamics that differ from the training set (Section 5).

## 2 RELATED WORK

Central to recent open-ended learning solutions is quality diversity (QD), based on NS, (Lehman & Stanley, 2011a), (Lehman & Stanley, 2011b) QD algorithms aim to return a diverse set of high-quality solutions by carefully balancing diversity and optimization (Pugh et al., 2016), (Mouret & Clune, 2015). In traditionally hard RL exploration problems Ecoffet et al. (2021) produced significant performance improvements by maintaining an archive of stepping stones from state to state.

QD itself however does not necessarily provide open-ended learning, most approaches for open-ended learning rely upon constantly generating new challenges for the agent to overcome. This can be through co-evolution (de Jong & Pollack, 2004), with an adversarial agent (Florensa et al., 2018a) or simply the agent itself (Team et al., 2021). In a similar vein research into self-generated curricula aims to automatically generate a sequence of objectives that efficiently allow the agent to learn complex behaviors. These approaches include: reverse goal-oriented curriculum generation (Florensa et al., 2018b), teacher-student curriculum learning (Matiisen et al., 2017), and procedural content generation (Justesen et al., 2018), (Hafner, 2021).

Another approach, *world-agent based co-evolution*, aims to constantly evolve the environment with the agent to generate pressure for agents to gather new skills. To facilitate open-ended learning we make use of a similar framework to POET (Wang et al., 2019) (Wang et al., 2020) where environment-agent pairs are evolved based on their QD using insights from the minimal criterion coevolution algorithm (MCC) (Brant & Stanley, 2017). The MCC performance ranking of agents in POET is based upon the agents total average episodic return from the extrinsic reward. Our approach extends this problem to the unsupervised domain, allowing for the discovery of diverse and useful sets of behaviors. PAIRED (Dennis et al., 2021) co-evolves 2 agents, a protagonist and antagonist where the protagonist aims to solve tasks generated by the antagonist. Regret is then defined as the average score of the protagonist and the best score of the antagonist over multiple trials. A separate adversary agent is then also trained, with the objective to minimax regret between the 2 agents by automatic environment generation. In contrast POET effectively maintains a population of minimax agents. We found the POET framework to be a suitable procedure to design a rewardless open ended learning algorithm. The results in PAIRED are inconclusive as to whether POETs population QD approach outperforms PAIREDs adversarial approach, we leave this investigation to future work.

Our methodology aims to extend open-ended learning to the unsupervised RL setting, in an effort to generate increasingly sophisticated predictable and diverse behaviors. Significant previous work in fixed environment unsupervised RL has focused on intrinsic motivation (Oudeyer, 2007), (Oudeyer et al., 2007), (Chentanez et al., 2005), with examples including: empowerment (Klyubin et al., 2005), (Mohamed & Rezende, 2015a), state count based exploration (Bellemare et al., 2016), curiosity driven exploration (Pathak et al., 2017), surprise maximization (Achiam & Sastry, 2017) and minimization (Berseth et al., 2021). Compared to intrinsic motivation *skill discovery* aims to explore a group of skills that can then be exploited by imitation learning (Schaal, 1997), inverse reinforcement learning (Russell, 1998), or fine-tuning. Skill discovery has generally made use of relations between RL and information theory (Ziebart et al., 2008), (Florensa et al., 2017), (Daniel et al., 2012), (Eysenbach et al., 2018), (Wang et al., 2018), (Sharma et al., 2020). DIAYN (Eysenbach et al., 2018) explicitly aimed to exploit HRL by designing a reward function to maximize state-entropy, so that skills are differentiated by visiting different state-space regions. DADS (Sharma et al., 2020) subsequently aimed to discover skills that are both diverse and *predictable*. Our work then develops the methodology in DADS to train agents over a range of increasingly complex environments, creating unending environmental pressure for agents to evolve ever more complex and diverse skills.

Gupta et al. (2020) provides another interesting approach to the problem of generalized unsupervised reinforcement learning by combining meta-learning (Finn et al., 2017) with DIAYN (Eysenbach et al., 2018) to automatically generate meta-training tasks. This is a useful approach for the fixed environment setting, but does not address the challenge of open-ended learning, only meta-training on a static state-transition distribution.

## 3    REWARDLESS OPEN-ENDED LEARNING (ROEL)

Building on previous work in unsupervised RL and mutual-information based exploration (Eysenbach et al., 2018), (Sharma et al., 2020), ROEL uses the POET framework (Wang et al., 2019) to endlessly generate agents capable of exhibiting increasingly complex, diverse and predictable behaviors. The policies learnt by ROEL can then be exploited to maximize a task specific objective using some form of supervision. Importantly our algorithms use of non-stationary environmental representations allows for the discovery of new emergent behaviors based on previous simpler behaviors.

### 3.1    POET FRAMEWORK

We use POET as a framework to endlessly discover agents with increasingly complex novel behaviors. POET accomplishes this through growing and optimizing a paired population of agents and unique environments (ea-pairs), contained within an environment-action list (ea-list). To control the evolution of the ea-list each environment is represented by an environmental encoding (EE), which maps environment attributes to an environment representation, and can easily be encoded and mutated. The Environmental Characterization (EC) is then used to describe the qualities of that environment, in ROEL we use the domain-general Performance of All Transferred Agents (PATA-EC)

from (Wang et al., 2020). The procedure to generate and search for behaviors within POETs main loop can then be summarized in the following steps:

1. **Mutate**: Every $M$ steps, provided an ea-pair exhibits suitable performance, it is mutated to a generate new child EEs. The children are then filtered via a *minimal criterion* (MC) (Brant & Stanley, 2017) to ensure the environment is not too easy or hard. EEs that satisfy the MC are then ranked based on the performance of the best agents and the most novel EEs added to the active population. When the active population reaches a predefined size the least novel pairs are dropped from the active environment and added to a set of archived environments. Novelty of newly-generated environments is based on how different the ranking of agents trained on the new environment changes compared to the ranking of agents on old environments, based on the idea that useful challenges should make novel distinctions amongst agents in a system (de Jong & Pollack, 2004).

2. **Optimize**: Every step POET optimizes every agent to maximize its reward on the given environment. In the original POET architecture this is the extrinsic reward from a Markov decision process, optimized with the Evolution Strategies (ES) algorithm (Salimans et al., 2017). In ROEL we optimize the mutual-information based reward as an RL optimization problem using Soft-Actor Critic (Haarnoja et al., 2018a).

3. **Transfer**: Every $N$ steps of POET's outer loop, the performance of all active agents is compared and ranked. If an agent on a different environment outperforms the current agent then the agent is replaced with the more performant agent. This constant transfer proved critical in POETs original implementation (Wang et al., 2019) for discovering stepping stones to more performant behaviors and to drive useful open-ended learning.

### 3.2 REWARDING ROEL

ROEL aims to create agents that exhibit a diverse set of predictable and useful behaviors across a range of problems. POET uses the extrinsic return generated from the agents interaction with the environment to generate an objective to minimize, as a supervised learning problem. The return of each ea-pair in POET is key to calculate each EEs PATA-EC and decide how to evolve the ea-pair. Subsequently in ROEL we use mutual information $\mathcal{I}$ to construct a reward function that aims to drive open-ended learning and motivate exploration, similar to Houthooft et al. (2017) and Mohamed & Rezende (2015b).

$$\mathcal{I}(X;Y) = \mathcal{H}(X) - \mathcal{H}(X|Y) \tag{1}$$

Following the definition of mutual information, shown in equation 1. Maximizing $\mathcal{I}$ with respect to $Y$ is equivalent to maximizing the entropy $\mathcal{H}(X)$ whilst minimizing conditional entropy $\mathcal{H}(X|Y)$, where $X$ and $Y$ are functions of state and actions respectively. This objective therefore aims to find actions to maximize state entropy, and is therefore exploratory as an RL policy. This property was recently used by Achiam et al. (2018), Eysenbach et al. (2018), Gregor et al. (2016) and Sharma et al. (2020) to learn a diverse and distinct set of skills in a policy.

In ROEL we make use of the reward function specified by DADS (Sharma et al., 2020) to learn a policy conditioned on a latent variable $z$, sampled from the transition dynamics prior $p(z) \sim \mathcal{Z}$. DADS differs from previous conditional entropy based approaches through the insight to leverage work in model based reinforcement learning (MBRL), where an objective is to learn a model to estimate the dynamics of the system $\hat{p}(s'|s, a)$. Therefore DADS learns 2 functions:

- A skill-conditioned policy $\pi(a|s; z)$ where skills are drawn from the larger skill set such that $z \in \mathcal{Z}$.
- *Skill-dynamics*: A skill-conditioned transition function $q_\phi(s'|s; z)$, that predicts the state transition given a skill $z$.

$$\mathcal{I}(s'; z|s) = \mathcal{H}(z|s) - \mathcal{H}(z|s', s) \tag{2}$$

DADS then uses the mutual-information between the next state with respect to a skill $z$, given the current state, to quantify how informative a skill $z$ is of the transition $s' \rightarrow s$, this is shown in

equation 2. Using a suitable approximation and the definition of conditional mutual information a reward function can be defined, as shown in equation 3 (full derivation found in Sharma et al. (2020)).

$$r_z(s, a, s') = \log \left( \frac{q_\phi(s'|s, z)}{\sum_{i=1}^{L} q_\phi(s'|s, z_i)} \right) + \log(L) \tag{3}$$

The reward $r_z$ in equation 3 encourages agents to exhibit skill diversity by sampling $L$ skills from the prior $z_i \sim p(z)$, to be used with the skill dynamics estimate $q_\phi$. We then use the return generated over several episodes to generate a statistic for the average return, that can be used to mutate and transfer agents within ROEL under the POET framework. This approach introduces non-stationarity into the skill transition and offers the following distinct advantages over simply training on a static environment.

*ROEL learns an open-ended diverse set of skills*

The combination of mutation and transfer learning was the key to POET learning increasingly complex skills in a supervised setting. For ROEL this procedure of gradually improving novelty via increased environmental complexity allows for simple, foundational, skills to be uncovered early in training. These are then used by ROEL as stepping stones to more complex skills when the challenge of more complex environments are presented (which in turn provide the opportunity to learn more complex behaviors). Generally this has the effect of improving the diversity of the set of skills uncovered as there is simply a larger distribution of possible state trajectories in more complex environments. But, designing building blocks to transfer the set of skills to the agent is key to efficiently discovering a large set of skills.

ROELs outer loop searches for environments that allow access to greater intrinsic reward, balancing novelty over performance via an MCC algorithm. This is implemented with a PATA-EC (Performance of All Transferred Agents Environmental Characterization) ranking, described in Wang et al. (2020), where each agent is ranked on an environment based on its performance against all other environments. EEs that change the ordering of other agents PATA-EC are favoured as this implies that the EE provides a new set of challenges to the set of active agents. To overcome this new challenge the agent will need to optimize different behaviors improving the diversity of skills learnt in the long run.

*ROEL is able to generalize over a large group of environments*

The search for novel environments in ROELs outer loop can also serve to improve the generalization of ROEL at test-time. By evolving EEs to generate novel behaviors ROEL is able to sample a large distribution of possible environments, but unlike domain randomization this search is directed to find environments that explicitly cause a change in behavior. This allows for ROEL to replicate the skills it learns across a much broader range of environments than if we were to simply train on a single-static environment. We explore this characteristic further in section 5.2 by comparing the performance of ROEL to DADS when tested on a hard EE unobserved at train-time.

## 4  EXPERIMENT SETUP

Using an implementation similar to DADS (Sharma et al., 2020), we use Soft Actor-Critic (SAC) (Haarnoja et al., 2019), an off-policy actor-critic deep RL algorithm, specifically EC-SAC, to train our policy $\pi(a|s, z)$. SAC (Haarnoja et al., 2018b) is based on the maximum entropy framework and uses entropy regularization to tradeoff robustness for exploration, as policies are encouraged to discover action-sequences with similar trajectories under a given skill.

To test ROEL we use the bipedal control environment (Brockman et al., 2016) with the custom EE extension described in Wang et al. (2019), allowing us to mutate the environment by varying a set of hand-picked obstacle hyper-parameters, such as stump shape, surface-roughness and gap width. This encoding offers sufficient opportunities for ROEL to discover diverse behaviors and is described in more detail in Appendix B. In future work we would like to investigate how much improving the

EEs expressiveness effects ROELs performance by using CPPNs (Compositional Pattern Producing Networks) (Stanley, 2007), similar to Wang et al. (2020).

For the experiments in section 5 we trained ROEL over 24 iterations (5400 episodic iterations) using a total of 224 processing cores distributed over 8 nodes (each with dual socket CPUs) for $\approx$ 12 hours. ROEL is highly parallel allowing for each ea-pair to be run asynchronously during the training step. Our baseline DADS environment is trained using the same parameters as the ROEL inner-loop, using the original bipedal normal environment. We find that ROEL takes slightly longer to train than DADS ($\approx$ 9 hours) as there is no need to calculate the PATA-EC and minimal criterion at the mutation and transfer steps. We also train a domain randomized (DR) DADS baseline on the parameters contained within the ROEL EC to test the capabilities of ROELs auto-curricula to generalize to unseen environments. We provide further details of our specific training-evaluation architecture and hyperparameters in Appendix A.

## 5 RESULTS

In this section, we present our results from using ROEL with the bipedal walker environment. Initially we qualitatively evaluate the types of skills learnt by ROEL and compare ROELs in and out-of-sample performance when used on a fixed skill $z$. We then quantitatively evaluate ROEL using DADS as a baseline, aiming to assess our initial hypothesis that ROELs use of open-endedness allows it to learn skills that are more diverse and general than unsupervised RL, trained on a single environment.

### 5.1 QUALITATIVE ANALYSIS

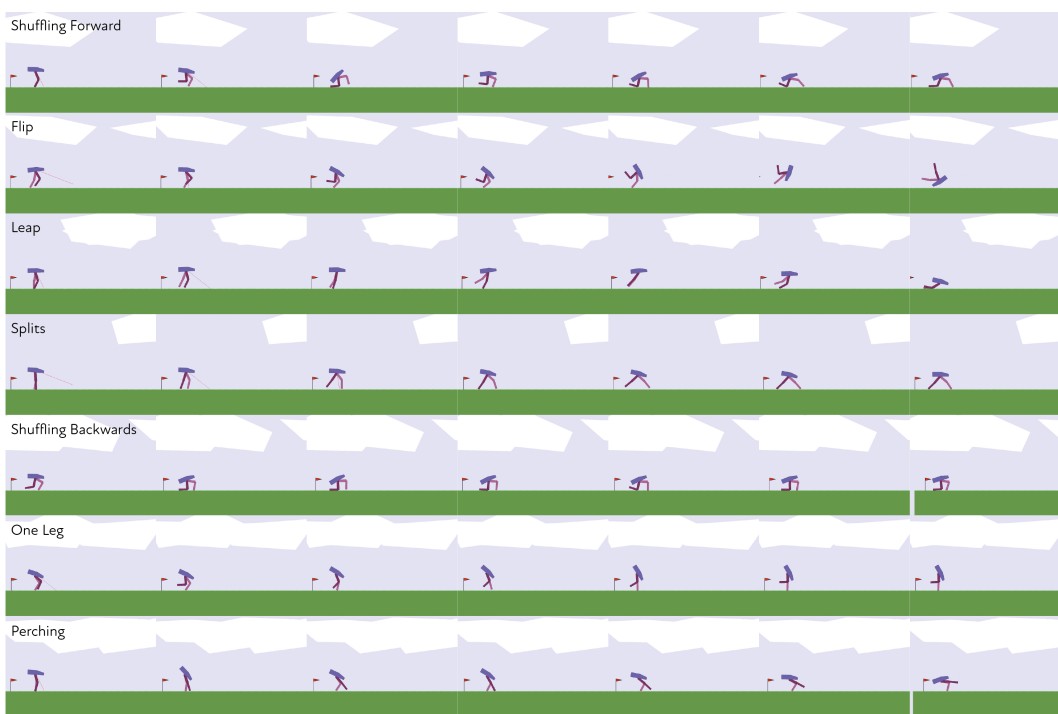

Figure 1: Examples of novel skills learnt by ROEL. Images are generated through policy rollout under a fixed skill $z$ specified at the beginning of each episode.

Figure 1 shows 7 of the 20 skills learnt by ROEL hand selected from a single run. Some of the skills are clearly motivated by the environments ROEL is exposed to during training. For example the *shuffling forward* skill allows the agent to move over pits generated in the training environment and keeps the LIDAR sensor close to the ground, helping to improve sensor accuracy. The *leap*

skill likely improves the intrinsic return when hills are present in the environment. The *one-leg* and *perching skills* may not immediately appear useful, but could prove key stepping stones to learn more complex modes of locomotion. For example the *one-leg* prior has learnt how to use the ungrounded leg to maintain balance for a short period.

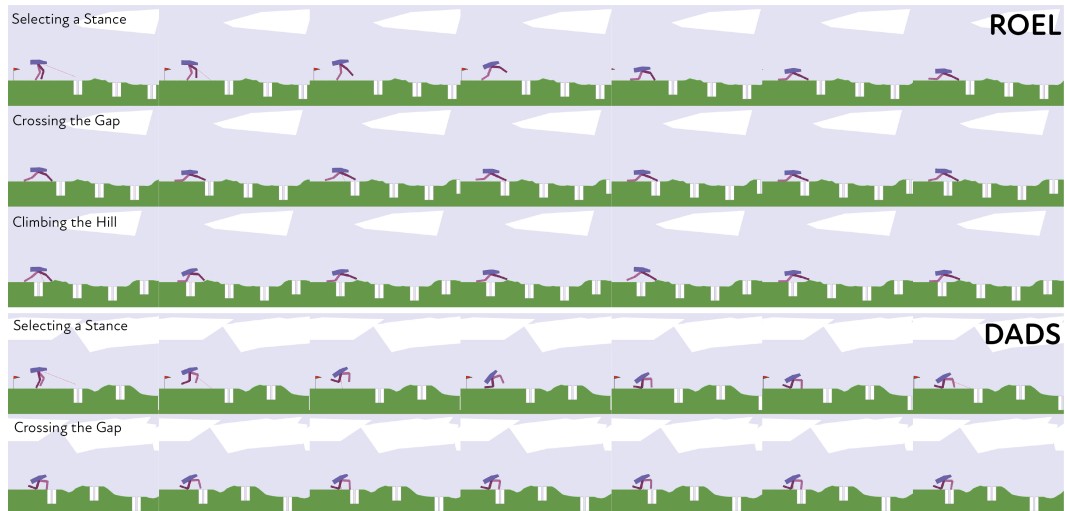

Figure 2: ROEL and DADS solving the same out-of-sample EE. ROEL is able to leverage its knowledge of previous tasks to rapidly cross a gap and climb the hill, whilst DADS meanders around the gap.

Figure 2 demonstrates how ROELs behavior differs from DADS in out-of-sample environments, using an EE containing gaps and roughness different from that seen by any agent at during training. When the ROEL agent in figure 2 crosses the pit it rapidly places a leg on the other side and pushes itself over the hole, it is then able to move its leg up to pull itself over the hill. The DADS agent on the other hand stops upon reaching the pit and explores the hole with its legs ground sensor, the agent then slowly shuffles over the hole, it never gets over the hill in the time horizon. We find this is a good qualitative example of how ROEL is able to leverage experience gained through mutation and goal-switching to respond to out-of-sample challenges.

## 5.2 QUANTITATIVE ANALYSIS

To assess the ability of ROEL to perform out-of-sample we use the EE seen in figure 2 and the original environmental encoding to generate samples under each skill prior.

It is difficult to directly measure the diversity of each agents prior based on samples from the environment. Observing diversity from the agents interaction with the environment alone requires us to characterize each with a suitable statistic. If the environment itself features stationary agents-environment dynamics, the variance of the trajectories can be used to measure behavioral diversity as the behavior is time-invariant. This is the case in Mujoco (Todorov et al., 2012) where the environment is a flat-plane: DADS therefore compares novelty via the agents state-transition. With bipedal, we cannot make this assumption, because the terrain is non-uniform and the agent-environment dynamics change with time. For ROEL, this shift in dynamics is also made intentionally via our constant evolution of the EE.

An ideal statistic for ROEL would characterize each skills policy dynamics, independent of the environment based on observations alone. We find that the state-space bayesian forecasting literature West & Harrison (1997) does offer some interesting possible solutions to this problem but this is still challenging. We would need to design a transfer function capable of capturing the skills fundamental dynamics but we find this to be outside the scope of this paper and leave this future work. Alternatively, we use the return (cumulative reward) from each episode to characterize each prior. This is based on the intuition that if the reward function itself describes an example of a novel

behavior, the difference in return of priors can be used as a heuristic to measure diversity. In bipedal, the reward function is based on the ability of the agent to reach the end of the episode.

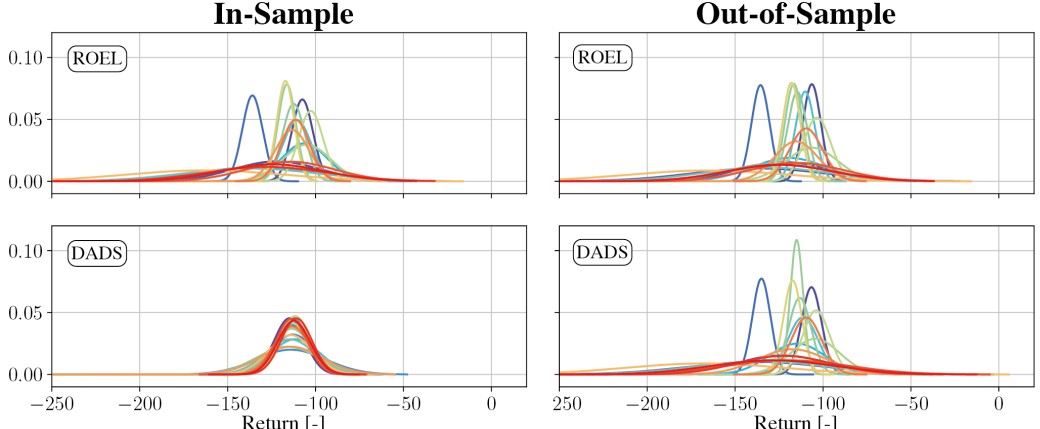

Figure 3: Dispersion of skill returns using the original bipedal extrinsic reward function. Each skill sample has been fitted to a normal distribution and assigned a unique color.

Figure 3 shows the fitted distribution of returns from each prior, both for ROEL and DADS, on the in- and out-of-sample environments. To fit the distribution we sample 100 episodes from each prior and calculate the empirical mean and variance of each to fit a normal distribution. In-sample, the prior of each sampled mean is more dispersed than in DADS. This suggests greater skill diversity, helping to explain the behavior we qualitatively observe in figure 1. Out-of-sample, the return of each prior in ROEL is generally unchanged, whilst DADS features a large change in dispersion. This indicates that the skills learnt by ROEL generalize well to this new out-of-sample challenge and highlights the importance of providing a diverse set of environments during training if we want to learn a diverse set of skills.

Another useful indicator of the capability of our agent to generalize to new environments is the error between the expected observation trajectory from the skill-dynamics network and the actual observation trajectory. If the skill-dynamics network is capable of accurately predicting the future observations, then the value of intrinsic motivation used to train agents will promote skills that are more diverse and predictable. The implementation of probabilistic ensembles with trajectory sampling (PETS) in Chua et al. (2018) suggests, that accurate state estimation is key for successful model based reinforcement learning, and adds a degree of confidence in our approach. The accuracy of our skill-dynamic network is therefore a useful heuristic for the performance of ROEL on downstream tasks.

Figure 4 plots the error in the prediction from the skill-dynamics network for both in- and out-of-sample EEs on the core hull dynamics. For our baseline, we instead used DADS trained with domain randomization (DR) across the same EE as ROEL. This provides a more accurate comparison of how our auto-curricula based approach in ROEL compares to a naive DR approach (when training without DR the error of DADS out-of-sample diverges to 100 times that of ROEL due to over-fitting). In-sample, we find DADS outperforms ROEL consistently with a smaller mean and variance on all 3 parameters.

Out-of-sample however, DADS often fails to predict the skill dynamics, rapidly diverging to an error 3 times that of ROEL in some cases whilst ROEL is generally unchanged. An exception to this is in the *y-velocity*, where the mean of ROEL and DADS-DR is generally the same, but DADS-DR has less variance. The *x-velocity* state offers some of the most interesting results, showing DADS-DR rapidly diverging in both mean and variance, whilst the error in ROEL is minimal. This difference can be explained by the differences in the DADS-DR and ROEL training procedures. In ROEL, the auto-curricula encourages longer episodes so that the behavior dynamics are longer. However, in DADS-DR, the environment selection is random and this means that the hardest obstacles often appear within the training environment so that the agent finds it difficult to traverse and this causes the episode to terminate earlier. This helps to explain why the *x-velocity* prediction is better, be-

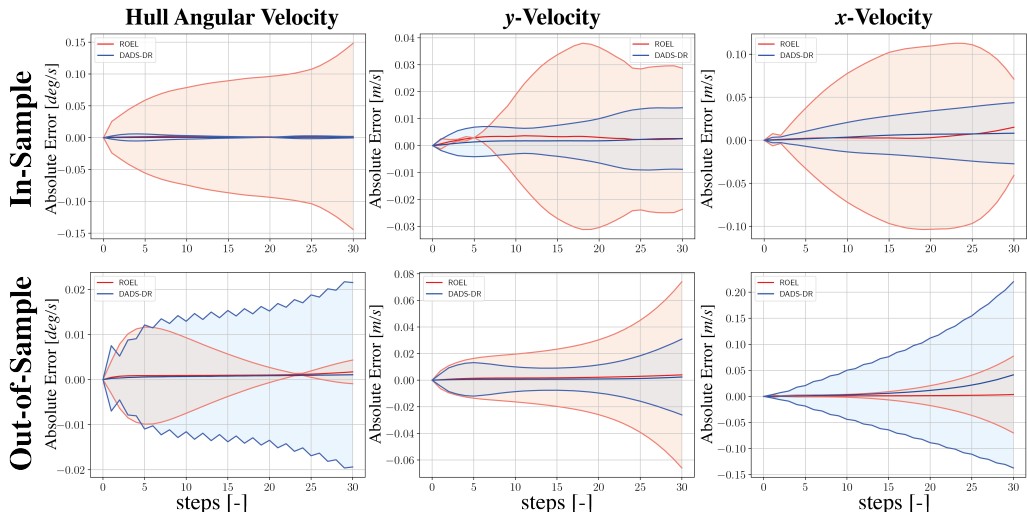

Figure 4: Comparison of the in and out-of-sample performance of the skill-dynamics network, sampled from 5 random discrete priors. The average error between the predicted and actual observation is plotted and enclosed by a shaded region out to 1-SD. Note the different y-axis scale in each graph.

cause ROEL is able to observe a more practical set of trajectories to learn generally diverse skills. Whilst the *y-velocity* prediction is relatively accurate for both ROEL and DADS-DR the reduced variance in ROEL is likely due to the environment observing a larger range of failures caused by the environment, which are more likely in the off-sample harder bipedal environment.

The fact that ROELs prediction error is unchanged in the out-of-sample environment is evidence that the skill-dynamics accurately generalize to unknown environments. Consequently, this suggests that the intrinsic reward received by ROEL during training captures the general environment dynamics. ROEL is therefore able to learn skills that generalize over a much larger distribution of possible environments. These skills are then pushed to be more diverse, as the outer loop of ROEL encourages the development of environments that lead to long term maximization of intrinsic reward leading to a set of novel, predictable and useful skills.

## 6 CONCLUSION

In this work we described how combining previous work in conditional entropy based unsupervised RL with POETs open-ended framework lead to the development of ROEL. An algorithm that aims to automatically generate increasingly complex and useful skills by generating its own curricula without the need for supervision. We demonstrated that this approach produces agents with skills that are more robust, when used in out-of-sample environments, than simply training on a single static environment.

The most compelling potential of this work lies in the open-ended nature of our methodology. Our skills are automatically generated by solving challenges presented by co-evolution, but as we do not need to specify an extrinsic reward function, provided a suitably expressive EC, ROEL should learn to generate skills indefinitely that constantly increase in complexity as the agent is presented with increasingly difficult challenges.

In future work, we aim to further investigate the open-endedness of ROEL, by training the agent for many more iterations and aiming to discover how diverse the learnt skills may become. Further, we plan to investigate how ROEL can be combined with CPPNs to generate a more expressive EC (Stanley, 2007), and how this could be directed to guide novel skill generation beyond the MCC criteria. Following this, we plan to investigate how ROELs learnt skills can be used in concert to exhibit more complex behaviors at test time, perhaps by simultaneously learning a meta-controller (Gupta et al., 2020) that searches for novel skill combinations.

STATEMENT OF REPRODUCIBILITY

To ensure reproducibility of the algorithm and results described within our paper we have included parameters of the setup used to train and evaluate ROEL in Appendix A, along with a general description of our implementation in section 4. After opening the discussion forums for all submitted papers we shall make a directed comment to the reviewers and area chairs with an anonymous repository containing:

- **Source Code** of our model, including methods to train, test and evaluate ROEL.
- **Videos of demonstrations** of the behavior of our trained agents.
- **Models and Datasets** used to evaluate the agents and generate figures in the report.

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

## A    IMPLEMENTATION DETAILS

In this appendix we provide details of our training setup for our DADS baseline and ROEL implementation.

### A.1    BASELINE

We use similar hyperparameters to the DADS implementation for our baseline. All skills are parameterized via a one-hot vector sampled from the uniform prior $p(z) = \frac{1}{D}$, where $D = 20$ is the number of skills, we only resample $z$ at the beginning of an episode. We use the EC-SAC implementation (Haarnoja et al., 2018a) to optimize the policy $\pi(a|s; z)$, based on the implementation found in TF-Agents Guadarrama et al. (2018). The SAC actor and critic networks both use 2 hidden layers with dimensions $(512, 512)$. The critic network is updated using Adam (Kingma & Ba, 2015) with a fixed learning rate $3 \times 10^{-4}$, trained over a fixed initial entropy coefficient $\alpha = 0.1$, updated via a soft update target function with co-efficient 0.005. Similar to DADS the policy is stochastic and parameterized on a normal distribution $\mathcal{N}(\mu(s, z), \sum(s, z))$, where $\Sigma$ is a diagonal covariance matrix that transforms the output to be in the range $[-1, +1]$ via a tanh function.

The skill-dynamics network uses the same dimensions $(512, 512)$ as the actor and critic networks. Our implementation uses a replay buffer to collect samples of reward and trajectory, with batch size of 256. The policy and critic networks are then updated via gradient descent every 500 steps for 64 steps of Adam.

### A.2    ROEL

Our ROEL implementation is built on-top of the DADS baseline and use the same architecture and hyper-parameters to calculate ROELs intrinsic reward. To train ROEL we maintain a list of 6 active ea-pairs ranked via their PATA-EC (Wang et al., 2020). Each active ea-pair is eligible to reproduce if it's average intrinsic reward is $> -0.1$, the child will then be considered for addition if its average intrinsic reward over 5 episodes on any active agent satisfies an MCC (Brant & Stanley, 2017) in the range $[0.05, 100]$. We found that a min MCC of $0.05$ provided frequent updates but prevented

Table 1: Bipedal walker observations

| Num | Observation | Min | Max | Units |
|---|---|---|---|---|
| 0 | Hull Angle | 0 | $2\pi$ | ° |
| 1 | Hull Angular Velocity | $-\infty$ | $+\infty$ | $°/s$ |
| 2 | Vel x | -1 | +1 | $m/s$ |
| 3 | Vel y | -1 | +1 | $m/s$ |
| 4 | Hip Joint 1 Angle | $-\infty$ | $+\infty$ | ° |
| 5 | Hip Joint 1 Speed | $-\infty$ | $+\infty$ | $°/s$ |
| 6 | Knee Joint 1 Angle | $-\infty$ | $+\infty$ | ° |
| 7 | Knee Joint 1 Speed | $-\infty$ | $+\infty$ | $°/s$ |
| 8 | Leg 1 Ground Contact Flag | FALSE | TRUE | - |
| 9 | Hip Joint 2 Angle | $-\infty$ | $+\infty$ | ° |
| 10 | Hip Joint 2 Speed | $-\infty$ | $+\infty$ | $°/s$ |
| 11 | Knee Joint 2 Angle | $-\infty$ | $+\infty$ | ° |
| 12 | Knee Joint 2 Speed | $-\infty$ | $+\infty$ | $°/s$ |
| 13 | Leg 2 Ground Contact Flag | FALSE | TRUE | - |
| 14-23 | 10x LIDAR Readings | $-\infty$ | $+\infty$ | $m$ |

Table 2: Bipedal walker actions

| Num | Name | Min | Max |
|---|---|---|---|
| 0 | Hip 1 | -1 | 1 |
| 1 | Knee 1 | -1 | 1 |
| 2 | Hip 2 | -1 | 1 |
| 3 | Knee 2 | -1 | 1 |

the worst environments, we did not investigate the effect of reducing the max criterion. The same parameters are used to re-rank based on PATA-EC and conduct transfer learning across the ea-list every 8 steps of ROEL.

We train each environment in the active ea-pairs list for 40 steps every ROEL iteration, each parent can only produce 3 offspring each reproduction step, which itself occurs every 3 steps of ROELs outer loop. Therefore, during the first 3 loops of ROEL we only have 3 children and for the rest of training the entire capacity of the ea-pairs list is full. This produces a total number of 5400 training steps on all ea-pairs using 24 ROEL iterations: $((3 \times 3 \times 40) + (3 \times 21 \times 40))$. The baseline is trained for 5600 steps, the closest value to 5400 when check-pointing every 400 iterations.

## B  BIPEDAL ENVIRONMENT

### B.1  WALKER

The bipedal walker environment used is an openAI gym environment (Brockman et al., 2016) with a continuous observation type Box(24) and continuos action space, type Box(4). Table 1 shows the 24 observations received by the agent at each state that we represent with $s$ in our definition of ROEL. Similarly, table 2 shows the 4 actions available to the walker. Figure 5 shows the states and actions labelled on the bipedal walker.

### B.2  EXTRINSIC REWARD

Algorithm 1 outlines the procedure used to obtain the extrinsic reward $r$ from each step of the bipedal walker environment. This is based principally on the distance from the start point in the horizontal $x$ direction and 4 motor action states, encoded as a tuple, $a = \langle a_{hip_1}, a_{hip_2}, a_{knee_1}, a_{knee_2} \rangle$. A reward shaping term ($\Delta$) is also included to encourage the agent to move forward and ensure the head is

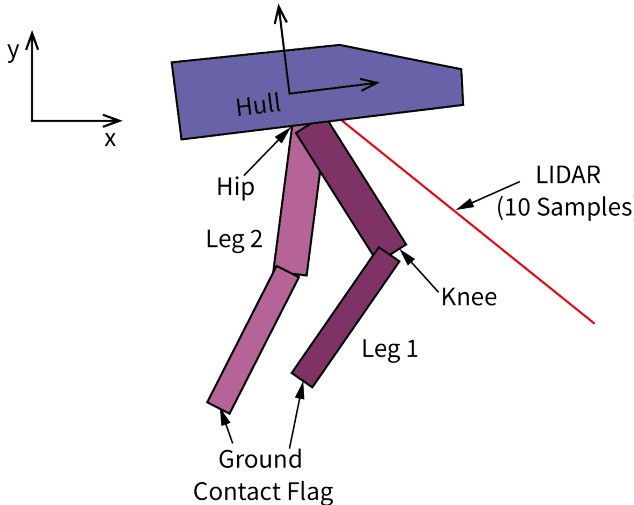

Figure 5: Components of the bipedal walker

kept straight. The episode terminates if it reaches the finish position ($x_{finish}$) or falls over and collides with the ground. All terms are normalized by a $scale$ variable to ensure reward behavior is generated correctly regardless of the rendering viewport size.

---

**Algorithm 1:** *Bipedal Walker Reward* reward function takes an input of hull $x$ position from the state-transition, actions $a$, and reward shaping term $\Delta_{previous}$

---

**Function** *reward($x$, $a$, $\Delta_{previous}$)* **is**

$\quad \Delta_{current} \leftarrow \frac{130x}{scale} - 5|x|$ ; $\qquad\qquad\qquad$ ▷ *Calculate reward shaping*

$\quad r \leftarrow \Delta_{previous} - \Delta_{current}$;

$\quad$ **for** $a_i$ *in* $a$ **do**

$\quad\quad | \quad r \leftarrow r - (0.00035\, \tau_{motor} \times clip(|a|, 0, 1))$ ; $\qquad\qquad$ ▷ *Slow motor rate*

$\quad$ **end**

$\quad$ **if** $fallen == True$ **then**

$\quad\quad | \quad r \leftarrow -100$ ; $\qquad\qquad\qquad\qquad\qquad\qquad$ ▷*Punish model if fallen over*

$\quad$ **end**

$\quad \Delta_{previous} \leftarrow \Delta_{current}$ ; $\qquad\qquad\qquad\qquad\qquad$ ▷ *Update reward shaping term*

$\quad$ **return** $r$, $\Delta_{previous}$

**end**

---

### B.3 ENVIRONMENTAL ENCODING (EE)

Our EE is inherited from the original POET bipedal implementation (Wang et al., 2019) and includes the following attributes:

- **Ground roughness**, controls how much the terrain moves up and down. ROEL sees a max of 1.0 at train time and our out-of-sample evaluation uses 5.0
- **Pit**, controls how large the pit gaps are in the environment, we use a large range at train time but the out-of-sample evaluation [0.5, 0.5] small gap is not seen during train time.
- **Stump**, controls how large stumps are made in the environment and are used in several train environments and are not used during the out-of-sample environment.
- **Stair**, adds stairs for the agent to traverse and are not used during training or out-of-sample.

The default EC used to start ROEL and train the DADS baseline sets all the EE variables to zero.

