# OpenReview forum: "Rewardless Open-Ended Learning (ROEL)"
_ICLR.cc/2022/Conference — ICLR 2022 Submitted_

### Official Review · Reviewer_kQnm · 2021-11-01

**Correctness:** 3
**Technical Novelty And Significance:** 3
**Empirical Novelty And Significance:** 2
**Recommendation:** 6
**Confidence:** 2

**Main Review:**

## Strengths
- The combination of open-ended and rewardless learning is very natural and synergistic -- it moves strongly in the direction of learning without hand-crafted structures.
- The algorithm is clearly presented and represents a paradigm with wide applicability.

## Weaknesses
- One of the main selling points of ROEL is that it is rewardless (unlike POET), which means that is should easily transfer to other environments. This is not evaluated, though I understand that the original version of POET is restricted to one environment.
- It is difficult to see a significant qualitative difference between ROEL and DADS as illustrated in Figure 2.
    A video would likely better serve this end.
- The quantitative analysis is primarily indirect; it supports the hypothesis but not in a definitive way.
- A comparison against POET would be particularly salient as it is the open-ended learning baseline.


## Recommendation
I recommend "accept", although I could be convinced either way upon further discussion.


## Justification
Although I point out a number of weaknesses with the evaluation, in particular, I believe the evaluation is an intrinsically difficult step for this type of work
The evaluation that is performed supports the claims of the paper, and the overall quality is still sufficient for acceptance.
Furthermore the novelty of the algorithm and its clear presentation lay a good foundation for future work.

Specific criteria:
- Correctness: 3
- Technical Novelty and Significance: 3
- Empirical Novelty and Significance: 2

## Questions
- What were the particular difficulties in combining these two algorithms/paradigms?


## Additional Comments
A caveat of my review is that while I am familiar with reinforcement learning, I am not up-to-date on the literature surrouding POET and DADS.
I can judge the work on its internal consistency and how much further it develops these two approaches, but if there are other similar approaches, I might not be aware of them.

- s1 p6: "We hypothesis" -> "We hypothesize"
- s4.1 Figure 2: This sequence of frames is not altogether convincing of the point being made. A video might be necessary.
- s4.1 p2: "this is a good qualitative example of how ROES is able to leverage experience gained through mutation & goal-switching" -- this claim does not seem adequately defended as I think the example only shows that ROEL might be more robust than DADS rather than specifically leveraging experience.
- s4.1 p2: "mutation & goal-switching" -> "mutation and goal-switching"
- s4.2 p2: "the reward function is based on the ability of the agent to reach the end of the episode" --
    Does reach the end of the episode mean that the agent moves rightward or simply that it does not fall before some number of timesteps?
- s4.2: "in sample" -> "in-sample", "out of sample" -> "out-of-sample"
- Figure 4: Highlight the fact that the in-sample plots have a highly zoomed-in y-axis (which is hard to read).
    I did not realize this at first and thought ROEL was somehow highly noisy compared to DADS in-sample, but in reality, the scale is too small to make a difference.
    Maybe it would just be better to keep the in-sample and out-of-sample plots at the same scale to demonstrate that ROEL is very similar to DADS in-sample, relatively speaking.
- s4.2: Probably beyond the scope of this paper (no expectation of implementing it), but a behavior/skill embedding space would be a great way to measure the diversity of behaviors, although this might be very difficult to implement.
- s4.2 last paragraph: "predictable & useful skills" -> "predictable and useful skills"


**Summary Of The Paper:**

Rewardless Open-Ended Learning (ROEL) is fundamentally a combination between two Paired Open-Ended Trailblazer (POET) and Dynamics-Aware Unsupervised Discovery of Skills (DADS).
POET presents a framework for "open-ended learning" which automatically generating progressively more difficult environments to elicit agents with novel behaviors.
DADS introduces a method of "unsupervised RL" which learns a variety of skills without rewards solely based on mutual information between state transitions and skill-determining latent variable.
The result is an RL framework which generates environments to elicit controllable behaviors without a reward function from the environment.

The following empirical evaluations are performed:
- Qualitative assessment of skills learned by ROEL
- Qualitative comparison or ROEL and DADS
- Measurement of rewards of different skills learned by ROEL and DADS
- Measurement of state predictability of skills in different environments by ROEL and DADS

The primary contributions of the paper are:
- Combining open-ended RL and unsupervised RL
- ROEL yields a more robust set of policies in a bipedal-walker environment than the DADS baseline.


**Summary Of The Review:**

ROEL is a logical and synergistic combination primarily of rewardless learning (DADS) and open-ended learning POET.
The algorithm has a clear applicability and a good presentation.
The experiments are reasonable and support the primarily claims of the paper, although the experiment measure indirect aspects of the claims as direct experimentation is intrinsically (to the claims) difficult.

---

> ### Author Response · Authors · 2021-11-16
> **Initial response to reviewer kQnm (part 1)**
>
> ## With respect to Weaknesses:
>
> 1. This is interesting: we did not consider direct transfer across completely different environments but only the out-of-sample environments of the bipedal environment. We would be interested to see how the approach generalizes to different agents. The comments of reviewer 2GeS about using ROEL in Mujoco would be an interesting experiment for this as we could investigate if ROEL is able to transfer skills across different agents dynamics. Whilst we do not directly address this here using skills learnt from other environments or even agents would certainly be a useful method to develop, both practically to reuse skills from other agents and to further develop the sub-field of open-ended RL.
>
> 2. We have now added videos to supplemental material.
>
> 3. We agree this a challenge and we note in the paper how we still lack a good statistic to compare different skills. We fear we may have to directly learn this, which is likely expensive and we describe this problem in detail in paragraph 3 of sec 4.2 on page 7 sec 4.2 quantitative analysis with the paragraph beginning “An ideal statistic for ROEL...”
>
> 4. We agree this would be useful. We think a challenge we still face in this area is proving open-endedness. For example in Enhanced POET, the authors note that the algorithm does begin to 'peter out’ after significant runs as the agent simply finds it difficult to learn any more complex environments. Whilst their EC is more expressive than what we use here, with CPPNs their use of a reward function could also be the limit. A true test of open-endedness would likely be to run ROEL on the enhanced POET CPPN environment for long enough to see if it is capable of generating the same set of behaviours as observed in POET with a reward function, and then exceeding them without petering out. Unfortunately, we believe this is computationally intractable using our own research facilities, as Enhanced POET was trained for 12 days on 750 CPU cores for 60,000 iterations and the change in ANNEC metric gradient only occurs after 30,000 iterations. Using the same compute used for ROEL would take approximately 40 days, which is much longer than our organization allows for a single job to be on the cluster. We think this is an important aspect to be addressed by researchers with access to larger compute facilities both in simulation and on real world robotic tasks.
>
> ## Questions:
>
> **What were the particular difficulties in combining these 2 algorithms/paradigms?**
>
> ### Implementation
> The authors of DADS provided a useful code base already written in python. We found it necessary to rewrite a large portion of the code using an OOP paradigm to allow us to manage multiple instances of the DADS agent, as the original implementation only considers a single agent and domain. This took considerable time and effort to rewrite because TensorFlow (TF) v1 is principally declarative so you must free the agent from memory and clear the keras graph backend each time a new ea-pair is evaluated. We wrote a procedure that allows for this, but believe that it could be improved if using a framework that allows for JIT compilation like TF v2, PyTorch or JAX. This was necessary because POET considers a group of paired environments. We also need to save each of these into persistent memory, as the TF v1 del agent operation means that agents cannot be stored in volatile memory as variables. We think that the load and unload operation of this naturally leads to a slow down.
>
> ### Experiment design
> This was and realistically still is a challenge and as you stated in your review is an intrinsically difficult step for this type of work. I do not think we have solved the problem of defining what truly makes an unsupervised approach better than another in this work as we can only observe the secondary results of behaviours either qualitatively from video demonstrations, or based on some external hand designed task specific objective such as the reward function. Ideally, we need a task agnostic metric to compare the performance of unsupervised groups of behaviours. We believe the MI criteria is one of the best heuristics we have for this, but believe further investigation would be useful.
>
> ### Environmental Characterisation (EC)
> In a more general sense, the design of the Environmental Characterisation (EC) is a key design decision and limits the utility of ROEL (and in fact of all POET type approaches). We do not use CPPNs in this paper, principally due to computational complexity of their approach, but the decision of whether to use a parametrized EC, as we do here, or a more generative EC from CPPNs or perhaps GANs, is still an open research question. We lay the foundations of a general unsupervised approach in this paper, but the question of how to apply an EC based on a downstream task is key to future research, especially if we wish to eventually make algorithms that are truly open-ended across a large variety of tasks.

---

> > ### Comment · Reviewer_kQnm · 2021-11-22
> > **Review overall unchanged**
> >
> > Thank you for the rebuttal.
> >
> > My review remains unchanged because I still think the major issue for the paper is the way that ROEL is evaluated.
> > I think there is a good attempt here, hence my 6, but I do not find the evaluation convincing enough to go higher.
> > I also think it would be good to explicitly include some of the wisdom gained from integrating the two approaches as this will bolster the contributions of the paper (especially in light of the difficulty of evaluation).

---

> ### Author Response · Authors · 2021-11-16
> **Initial response to reviewer kQnm (part 2)**
>
> ## Additional Comments:
>
> - s1 p6: "We hypothesis" -> "We hypothesize" corrected.
>
> - s4.1 Figure 2: A collection of videos have now been added.
>
> - s4.1 p2: This is more related to the mechanics of how ROEL learns during training, as the ea-pair ranking aims to transfer agents to new environments to learn skills which changes the agent-environment dynamics and forces the DADS part of ROEL to learn a more diverse set of skills to ensure predictability. The example in figure 2 is used as an indicator of the utility of this mechanism, which we infer allows for improved robustness as shown in figure 4 when the skill-dynamics offers significantly improved OOD performance over the baseline.
>
> - s4.1 p2: "mutation & goal-switching" -> "mutation and goal-switching" corrected.
>
> - s4.2 p2: Yes the hand crafted reward function encodes an objective to move further to the right, but the episode terminates if the agent collides with the ground. This means the agent is encouraged to stay upright so that it can maximize return and move to the right. We have added a definition of the reward function in appendix B.2 page 16.
>
> - s4.2: "in sample" -> "in-sample", "out of sample" -> "out-of-sample" corrected.
>
> - Figure 4: We have added a note on the figure definition and made the font larger, we have kept the y-axis scale different as we think it is still interesting to see that the DR-DADS agent outperforms ROEL on-sample. Also now we compare on Domain Randomization over the single environment baseline the size of the divergence in expected state error is less.
>
> - s4.2: This is an interesting idea. We read 'Learning an Embedding Space for Transferable Robot Skills' Hausman et al. Is this inline with the type of implementation you envision? We are unsure if we can implement this in time for the amendment deadline. However, I think measuring the diversity of behaviours is a key challenge to the experimental comparison of our procedure and somewhat limits progress in this area.
>
> - s4.2 last paragraph: "predictable & useful skills" -> "predictable and useful skills" corrected.

---

### Official Review · Reviewer_Bne6 · 2021-11-02

**Correctness:** 3
**Technical Novelty And Significance:** 3
**Empirical Novelty And Significance:** 2
**Recommendation:** 3
**Confidence:** 4

**Main Review:**

### Strengths
- Open ended learning and the unsupervised discovery of diverse skills is a very interesting problem and the paper suggests an extension of prior work to this intersection of approaches.
- The paper is well written, and presents a complicated approach clearly.


### Concerns

The evaluations of the presented algorithm are unconvincing.

- Primarily, the baseline is an algorithm trained on a single environment.  It is not surprising that a method trained on multiple environments (co-evolved or not) will generalize better to unseen environments.  If the contribution is meant to highlight the improved performance of co-evolution over a static distribution of tasks (or some other naive multi-task method), it would be nice to see that comparison.  The differences observed in Figures 3 and 4 could reasonably be explained by single vs multiple environment training, rather than the POET-like curriculum learned through ROEL.

- Second, the quantitative evaluation of unsupervised learning methods is a difficult problem in itself, but it would be nice to see some evaluation of fine-tuning to a particular “downstream” task with extrinsic reward (again, comparing to sensible baselines).

Without proper experimental comparisons, the significance and impact of the ROEL algorithm is difficult to assess.

## Post rebuttal update

Thank you for the response.  The evaluation against DR is a step in the right direction.  However, the evaluation still needs to be more mature (see my second point above) before I would be comfortable accepting this paper.  I encourage you to keep working on it and submit to another conference.

**Summary Of The Paper:**

This paper presents a method for learning skills in an open-ended, reward-free setting using coevolution of agents and environments trained to maximize the mutual information between a skill latent code and state transition.

**Summary Of The Review:**

Although the method is interesting and tackles a relevant problem, the experimental evaluations are not mature enough to provide a clear indication of the method's merit.

---

> ### Author Response · Authors · 2021-11-16
> **Initial response to reviewer Bne6**
>
> With respect to your Concerns:
>
> 1. Thanks for this comment. We agree with this and now compare to domain randomization for the prediction error in figure 4. This offers improved performance. We left the graphs in figure 3 unchanged as we believe this is an interesting illustration of how important environment design is for diverse skill discovery.
> 2. Yes. We agree a success metric would be useful to help understand downstream performance and are working on using the goal-conditioned approach described in the initial DADS paper to compare our approach to the baseline. As we also stated in our response to reviewer 2GeS, we believe that given we demonstrate improved OOP prediction with ROEL, if deployed to downstream goal-reaching tasks using PETS MBRL will demonstrate improved performance due to PETS (or MBRL in generals) high-dependency on accurate agent-environment prediction.

---

### Official Review · Reviewer_2GeS · 2021-11-04

**Correctness:** 3
**Technical Novelty And Significance:** 2
**Empirical Novelty And Significance:** 2
**Recommendation:** 3
**Confidence:** 4

**Main Review:**

Strengths

1. Relevance of problem setting -
The paper considers the very important setting of reward-free learning, since it is difficult to specify reward functions that align with our intended objectives and further cumbersome to provide rewards for real systems. The paper considers leveraging open-ended learning to try to learn a more diverse set of skills, since automatically generating challenging environments can enable new skills to be discovered.

Weaknesses

1. Experimental Evaluation  - missing evaluation on DADs environments

The proposed method can be viewed as DADs, augmented with a scheme to train on autonomously generated diverse environments (from POET). In the experiment section, the authors compare the performance of ROEL to that of DADs on held-out environment variations, and argue that ROEL is superior. However, they only use the bipedal walker which has very low action space (only 4 dimensions), and don’t test on any of the environments used in the DADs papers [2,3] which include control of much more complex agents such as the gym-ant, humanoid or the d-kitty robot. Since the skill-discovery from DADs has been shown to enable effective control in these agents, we need to evaluate if open-ended learning can scale to these agents as well. This is quite critical for general adoption of open-ended learning in unsupervised learning.

2. Experimental Evaluation - issues with current evaluation

For the only included environment (bipedal-walker), performance comparison on the out of distribution environment would be much more compelling if the authors included quantitative measures of performance such as a success metric. Instead the authors compare performance by showing visualization of 1 run for each method (Fig.2), but this could be susceptible to noise, it’s unclear if this behavior can be reliably replicated, and further from the visualizations, it’s unclear if the ROEL agent actually cross the gap successfully.

The authors do include some quantitative analysis of the prediction error of the observation trajectory from the skill network (Fig. 4), and show that DADs skill network overfits. But this is an argument for ROEL discovering more diverse skills, not whether these skills actually enable better performance on a given OOD environment, for which success metrics should be used.

In Fig.3 the authors plot returns from behavior resulting from different skill priors, and find that the return plot for ROEL is more diverse. They argue that diverse returns can be used as a heuristic to imply the corresponding skills are diverse, but I’m not sure this is necessarily true. For example, different skill priors could essentially learn to perform the same behavior but to varying degrees of proficiency, thus resulting in a diverse return plot without having skills that are really diverse.

3. Novelty

The novelty of the approach is very limited, as it essentially takes the POET algorithm for open-ended learning, and replaces the external environment reward with the exact reward used in DADs. It would have been interesting to consider how the DADs skill conditioned dynamics model that is learned could be used for ranking or selecting environments/agents in the open-ended learning part.


**Summary Of The Paper:**

This paper proposes combining unsupervised skill-discovery with open ended learning, where the environment and agents are continually evolved together. Specifically the proposed method (ROEL) is a combination of POET[1] with the intrinsic reward for skill-discovery in DADS[2], and is evaluated on a simulated bipedal walker environment.


**Summary Of The Review:**

While the premise of generating new environments to enable more diverse skills to be learned is intriguing, the experiments do not convincingly demonstrate this. The paper lacks evaluation on more complex environments where DADs is known to work well, and also doesn’t include quantitate performance metrics for the simpler bipedal walker considered.

---

> ### Author Response · Authors · 2021-11-16
> **Initial response to reviewer 2GeS**
>
> With respect to Weaknesses:
>
>
> ### 1. Experimental Evaluation - missing evaluation on DADS environments
>
> We agree that comparing to generalization of more complex action spaces is a useful area of research. However, we do not necessarily believe that this is required to prove the utility of ROEL. Whilst the action space of bipedal is simpler than Mujoco, the bipedal gym environment itself is comparatively more complex than the standard Mujoco benchmark (where the environment is a flat plane). We argue that whilst the dynamics of bipedal are indeed simpler than Mujoco this does not limit the agent’s ability to learn complex behaviours. This was proven in the original POET and enhanced POET paper, where despite using a large number of episodes for training the agent never converged on the tasks environmental characterization (EC) and demonstrated diverse behaviour to maximize return.
>
> For Mujoco the design of a suitable EC for training is not obvious, as we would need to consider a 3D EC, that as far as we are aware has only been studied in a limited setting [Lee et al 2020 Learning Quadrupedal Locomotion over Challenging Terrain] on a parametrized environment. We suspect CPPNs as used in enhanced-POET would provide a useful EC representation, but believe that the implementation and training of this is best left to future work and does not undermine the usefulness of ROEL. We could also vary the vehicle dynamics itself to enable skill discovery, and use a hierarchy of priors on downstream tasks in Mujoco.
>
> We believe that the need to define ECs for ROEL is probably one of its major limitations and is an area that we still do not have a good way to generalize across many environments, especially if we want to ensure open-ended learning can be achieved.
>
> ### 2. Experimental Evaluation - issues with current evaluation
>
> We have included a video of both agents on the complex environment gap task to help demonstrate improved performance across the OOD environment. This is in the file named off_sample_comp.gif the first is the DADS agent the second ROEL.
>
> We are working on demonstrating goal-conditioned RL for downstream tasks using the MPPI approach in DADS to reach a goal-conditioned state and we hope to add this before the deadline.
>
> Regardless, we believe that the improved generalisation of the skill-planner to OOD environments is a good indicator that our method will outperform the baseline on downstream tasks. Improved predictability of the agent-environment dynamics is key to MBRL algorithms such as probabilistic ensembles with trajectory sampling (PETS) [Deep Reinforcement Learning in a Handful of Trials using Probabilistic Dynamics Models Chua et al 2018] performance. Given that the results in figure 4 demonstrate that our skill-dynamic planner significantly outperforms the baseline planner, we are confident that ROEL will outperform the baseline when used with MBRL on downstream tasks.
>
> Yes, we agree that using the return of the original baseline is not an ideal heuristic for diversity as we are still comparing to a handcrafted objective function that is unlikely to capture the true diversity of the behaviour space. We ideally need a way to compare the trajectories of rollouts themselves, agnostic of a reward function. We do not know of a way to form such a statistic of the agents underlying dynamics, and we summarize this problem in s4.2p3.
>
> ### 3. Novelty
>
> Whilst we agree that ROEL essentially combines 2 existing algorithms, this is an area that is still in its infancy and an important problem area for us to address if we wish to generate algorithms that are truly open-ended. We believe that developing ROEL was an important first step in combining open-ended learning and unsupervised learning as reward functions fundamentally limit our capability to discover behaviours in a truly open-ended fashion.

---

### Official Review · Reviewer_HJjY · 2021-11-05

**Correctness:** 3
**Technical Novelty And Significance:** 2
**Empirical Novelty And Significance:** Not applicable
**Recommendation:** 3
**Confidence:** 3

**Main Review:**

## Pros:

1. The paper addresses an important problem: unsupervised skill discovery in (kind of) open-ended RL environments. This is exactly the kind of problem where specifying a concrete reward limit what can be achieved in terms of skill discovery. To me, the problem is real and practical.
2. The proposed framework of skill discovery in open-ended domains is novel, although combines two existing methods with each other.
3. This paper provides comprehensive experiments, including both qualitative analysis and quantitative results, to show the effectiveness of the proposed method.

## Cons:

1. I don't quite understand how exactly was the DADS baseline trained. The paper says it was trained "using the original bipedal default environment". There are two versions of the walker environment, according to its OpenAI gym implementation, namely normal (with slightly uneven terrain) and hardcore (ladders, stumps, pitfalls). Which one was it trained on? If DADS is trained on a single environment, it is not really fair to compare it with one trained on multiple environments for generalisation purposes.
2. Have authors tried training DADS using Domain Randomization (DR), where the training environment is sampled from a parameterized distribution, i.e., randomly sampling hand-picked obstacle hyperparameters, such as stump shape, surface roughness and gap width? The authors should show that ROEL works better than DADS trained on DR, otherwise, it is hard to argue about the effectiveness of the method.
3. I'm not convinced with the statement in Section 4.1 claiming that ROEL "outperforms" DADS in off-sample environments. What does "outperform" even mean in this particular experiment and how is it measured? All I see is a few cherry-picked environments where some skills of ROEL were able to pass through an obstacle where some skills of DADS weren't able to do this. First, I don't understand how the skills are selected in these experiments. How is z chosen? Have you tried all z's? Have you tried more example environments? Please elaborate on exactly what is going on here.
4. The structure of the paper can also be improved. Particularly, it is a bit unusual to see Implementation details (Section 3.3) in what seems like the Method (Section 3). The authors could consider adding an Experimental Setting section (after section 3) where such details will be described. Also, various future directions for work are described throughout the paper. Moving all those ideas to the Conclusion would improve the readability of the paper.

## Minor issues

- In PAIRED, a separate adversary generates the environments for the protagonist and antagonist. The antagonist doesn't do this.

## Additional issues

The issues below did not influence my score, but please address them in the updated version of the manuscript.

### Citations format:
- There are two basic citations commands in the `natbib` package
    - `\citet` for **textual** citations. This prints *Hinton et al. (2006)*
    - `\citep` for **parenthetical** citations. This prints *(Hinton et al.,* 2006*).*
- In the entire paper, authors make use of the `\citet` command (or generic `\cite` which can be problematic, see [natbib documentation](https://ctan.org/pkg/natbib), page 6). However, the majority of them should actually use the `\citep` command. According to the [ICLR 2022 formatting instructions](https://github.com/ICLR/Master-Template/raw/master/archive/iclr2022.zip):
    - "Citations within the text should be based on the natbib package and include the authors’ last names and year (with the “et al.” construct for more than two authors). When the authors or the publication are included in the sentence, the citation should not be in parenthesis using `\citet{}` (as in “See Hinton et al. (2006) for more information.”). Otherwise, the citation should be in parenthesis using `\citep{}` (as in “Deep learning shows promise to make progress towards AI (Bengio & LeCun, 2007).”). "

### Academic language and abbreviations:
- Please DO NOT use "&" instead of "and", "w/" instead of "with", etc, throughout the paper

### Punctuation errors:
- There are large number punctuation mistakes in sentences all over the paper. Especially, commas are very often missing. Please fix these.

# Post-rebuttal update

I thank the authors for their response and improving the manuscript based on my suggestions. Specially, I appreciate the authors running DADS on domain randomised version of the bipedal walker environment and making cosmetic changes to the paper.

However, after taking a look at the current version of the paper, I can't yet recommend that it is accepted in its current form. It seems like the qualitative and some quantitative analysis is still compared against DADS on the normal version of the bipedal environment. The performance gains against this baseline are expected and the comparison is unfair, as far as I'm concerned.

I therefore recommend that the authors continue improving the paper and submit it for another review process at one of the upcoming conferences.

**Summary Of The Paper:**

In this work, the authors a method that lies in the intersection of two subfields of RL, namely open-endedness and unsupervised skill discovery. Specifically, they introduce a method called Rewardless Open-Ended Learning (ROEL) which extends a recent method called POET to perform skill discovery in a reward-free setting rather than traditional supervised RL. To this end, they use mutual-information-based skill discovery techniques which have recently become a popular approach for unsupervised RL. The authors empirically demonstrate that ROELD is able to learn identifiable skills in bipedal walker environment.

**Summary Of The Review:**

While I think the paper addresses an important problem, I have major concerns about the claims the authors make given the experiments they have. Specifically, I've raised questions on the choice of baseline agents and comparison with them. Furthermore, the writing can be improved significantly.

---

> ### Author Response · Authors · 2021-11-16
> **Initial response to reviewer HJjY**
>
> ## With respect to your Cons:
> 1. We trained on the normal environment. This has now been corrected in the paper and we agree that training on a single environment may well be an unfair comparison.
>
> 2. We think the idea of using DR to compare is an excellent idea, and have retrained the DADS baseline on a DR environment to compare with the performance of ROEL. As suspected, this outperforms the static DADS baseline in out-of-sample tasks. However, we find that the state prediction error is still better using ROEL. We believe this is due to the auto-curricula of ROEL, as the DR approach often ends up with trajectories that are short because the obstacles presented are too difficult to learn useful behaviours. We believe this is a problem that will worsen as the domain is made more complex causing required behaviours to be more complicated to solve challenges presented by ROEL.
>
> 3. We agree ‘outperforms’ was a poor choice of words. We found the 2 demonstrations to be a useful qualitative comparison, by looking at the 2 agents with the greatest extrinsic reward, and observing how their behaviours differ when they meet unseen objectives. We train z's from a discrete one hot vector. Appendix A.1 provides a little more detail, but when evaluating this gives 20 skills to select from. These were then hand-picked to demonstrate useful behaviours within the paper. We have added a video of all of the skills in supplemental material that should give a better overview of all 20 priors. Some of the skills do not appear to generate useful behaviours at all, we believe to some extent this is a limitation of the maximum entropy methodology of DADS, as you effectively need to learn useless skill to learn good ones. Some useful further experiments to solve this could be:
>
>     - To train ROEL for longer, if we had a more open-ended Environmental Encoding it may be possible for the discrete skill
>     grouping to become saturated and all of the skills to behave as useful distinct behaviours. This is due to the transfer learning
>     step of ROEL.
>
>     - To change the ROEL evolution metric to be on goal states rather than on the DADS mutual information. This could help with
>     the problem of extracting useful downstream tasks at test-time too.
>
> 4. Thanks for this we agree that this is a good idea and have restructured the paper to reflect the comment by:
>     - Changed s3.3 to s4, named Experiment Setup and s5 to simply be called Results.
>     - We have left s5.2p2 and s5.2p3 in the quantitative analysis section, as we believe explaining the limitations of evaluating
>     diversity is necessary to understand the results within figure 3 and 4.
>
> ## Minor issues
>
> Changed s2p3 'The objective then becomes to minimax regret between the 2 agents' to -> 'A separate adversary agent is then trained with the objective to minimax regret between the 2 agents by automatic environment generation'.
>
> ## Additional issues
>
> ### Citations format:
>
> Thanks for letting us know. We have now corrected this and were not aware of this formatting difference.
>
> ### Academic language & abbreviations:
>
> We believe we have corrected these grammatical mistakes and changed several '&' -> 'and'.

---

> > ### Comment · Reviewer_HJjY · 2021-11-26
> > **Response to the rebuttal**
> >
> > I thank the authors for their response and improving the manuscript based on my suggestions. Specially, I appreciate the authors running DADS on domain randomised version of the bipedal walker environment and making cosmetic changes to the paper.
> >
> > However, after taking a look at the current version of the paper, I can't yet recommend that it is accepted in its current form. It seems like the qualitative and some quantitative analysis is still compared against DADS on the normal version of the bipedal environment. The performance gains against this baseline are expected and the comparison is unfair, as far as I'm concerned.
> >
> > I therefore recommend that the authors continue improving the paper and submit it for another review process at one of the upcoming conferences.

---

### Author Response · Authors · 2021-11-16
**For the attention of all reviewers**

Thank you to all for your detailed reviews and insightful feedback. We are delighted that you find our work interesting and useful.


**For your information:**
* We have attached an updated version of our paper along with supplemental video information.
* We are currently checking through our code to ensure it is anonymous and shall provide a link to download the repository and our trained models shortly.

---

> ### Author Response · Authors · 2021-11-18
> **Addition of repository**
>
> - If you would like to view the code we used to train ROEL please use the following link to download the code and train the agent, instructions to get started are contained within the README.md file.
>
> https://drive.google.com/drive/folders/1kaeE3JPm7U2eGHmCyyWALuW1WwdHJF1J?usp=sharing
>
> - We have also made a minor modification to the report PDF to ensure we remain below 9 pages in the main body of the report.

---

### Decision · Program_Chairs · 2022-01-20

**Decision:**

Reject

**Comment:**

I thank the authors for their submission and active participation in the discussions. This papers is borderline with the majority of three reviewers leaning towards rejection and one leaning towards acceptance. All four reviewers unanimously agree on the empirical validation being a major weakness of this paper with reviewer kQnm putting less emphasis on this shortcoming. Overall, I side with reviewers Bne6, 2GeS and HJjY, and believe the paper needs to a more thorough set of experiments. I therefore recommend rejection.